# Laryngopharyngeal reflux, gastroesophageal reflux and dental disorders: A systematic review

Jerome R. Lechien[1,2]*, Carlos M. Chiesa-Estomba[1,3], Christian Calvo Henriquez[1,4], Francois Mouawad[5], Cyrielle Ristagno[6], Maria Rosaria Barillari[1,7], Antonio Schindler[1,8], Andrea Nacci[1,9], Cyril Bouland[1,10], Luigi Laino[11], Sven Saussez[12,13]

1 Laryngopharyngeal Reflux Study Group of Young-Otolaryngologists of the International Federations of Oto-rhino-laryngological Societies (YO-IFOS), 2 Department of Otolaryngology-Head and Neck Surgery, Foch Hospital, School of Medicine, University Paris Saclay, Paris, France, 3 Department of Otorhinolaryngology—Head & Neck Surgery, Hospital Universitario Donostia, San Sebastian, Spain, 4 Department of Otorhinolaryngology and Head and Neck Surgery, Hospital Complex of Santiago de Compostela, Santiago de Compostela, Spain, 5 Department of Otorhinolaryngology and Head and Neck Surgery, CHRU de Lille, Lille, France, 6 Edwards-Realty, Mons, Belgium, 7 Division of Phoniatrics and Audiology, Department of Mental and Physical Health and Preventive Medicine, University of Naples SUN, Naples, Italy, 8 Department of Biomedical and Clinical Sciences, Phoniatric Unit, L. Sacco Hospital, University of Milan, Milan, Italy, 9 ENT Audiology and Phoniatric Unit, University of Pisa, Pisa, Italy, 10 Department of Stomatology-Maxillofacial Surgery, CHU Saint-Pierre, Brussels, Belgium, 11 Multidisciplinary Department of Medical-Surgical and Dental Specialities, L. Vanvitelli University, Napoli, Italy, 12 Department of Otorhinolaryngology and Head and Neck Surgery, CHU de Bruxelles, CHU Saint-Pierre, School of Medicine, Université Libre de Bruxelles, Brussels, Belgium, 13 Department of Human Anatomy and Experimental Oncology, Faculty of Medicine, UMONS Research Institute for Health Sciences and Technology, University of Mons (UMons), Mons, Belgium

* Jerome.Lechien@umons.ac.be

**Data Availability Statement:** All relevant data are within the manuscript and its Supporting Information files.

## Abstract

### Objectives

To investigate the role of gastroesophageal reflux disease (GERD) and laryngopharyngeal reflux (LPR) in the development of dental disorders.

### Methods

The first outcome was review of the role of reflux in the development of dental disorders in adults. The second outcome was review of the potential pathophysiological mechanisms underlying the association between reflux and dental disorders. Three investigators screened publications for eligibility and exclusion based on predetermined criteria through a literature search conducted on PubMed, Cochrane Library, and Scopus according to the Preferred Reporting Items for Systematic Reviews and Meta-Analyses (PRISMA).

### Results

From 386 publications, 24 studies were kept for analysis. Objective approaches were used in 16 studies to confirm GERD diagnosis. Pharyngeal reflux episodes (LPR) were considered in 2 studies. No study considered nonacid reflux. The study results supported a higher prevalence of dental erosion and caries in reflux patients compared with healthy individuals.

**Funding:** The authors received no specific funding for this work.

**Competing interests:** The authors declare that they have no conflict of interest.

Patients with dental erosion have a higher prevalence of reflux than controls. The pathophysiological mechanisms would involve changes in the saliva physiology. No study investigated the microbiota modifications related to reflux although the findings are supporting the critical role of microbiota change in the development of dental disorders. There is an important heterogeneity between studies about diagnostic methods and clinical outcome evaluation.

## Conclusion

The involvement of reflux in the development of dental disorders is not formally demonstrated and requires future investigations considering pharyngeal acid and nonacid reflux episodes and in particular their potential impact on oral microbiota.

## Introduction

Laryngopharyngeal reflux (LPR) is an inflammatory condition of the upper aerodigestive tract tissues related to direct and indirect effect of gastroduodenal content reflux, which induces morphological changes in the upper aerodigestive tract [1]. The incidence of LPR-associated symptoms ranges from 10 to 30% of people of Western countries [2,3] and would be increasing concerning the changes in a modern lifestyle and dietary habits [4]. The LPR is involved in the development of many otolaryngological diseases through the deposit of gastroduodenal enzymes into the mucosa of the upper aerodigestive tract. Thus, pepsin has been identified in the laryngeal [5], hypopharyngeal [6], oral [7], nasal [7], tears [8], and Eustachian mucosa as well as in the secretion of chronic rhinosinusites [9] and chronic media otitis [10]. The involvement of reflux in the development of dental disorders has been suspected for several decades. Reflux would be responsible for increasing risk of oral mucosa inflammation [11], dental caries [11] and erosion [12]. Nowadays, the pathophysiological mechanisms underlying the development of dental disorders related to reflux are still poorly understood. Many hypotheses have been proposed including the reduction of the salivary buffering capacity or the modification of the pharyngeal/oral microbiota by acid reflux episodes [13].

The aim of this paper is to review the current literature about the role of reflux in the development of the following dental disorders: mucosa inflammation, dental erosion and caries.

## Materials and methods

The criteria for considering studies for the systematic review were based on the population, intervention, comparison, and outcome (PICO) framework [14]. The review was conducted regarding the PRISMA checklist for systematic reviews [15].

### Types of studies

The studies were included if they investigated the association between reflux and dental disorders (i.e., mucosa inflammation, dental erosion and caries) through clinical prospective, retrospective, randomized or non-randomized studies, or basic science research published in English or French in peer-reviewed journals. We considered the studies conducted in an adult population. Only studies reporting data for more than 10 patients were considered.

## Participants, inclusion/exclusion criteria

Papers were included for analysis if they reported the diagnostic method for reflux disease (LPR or gastroesophageal reflux disease (GERD)). The clinical papers were included if the authors attempted a rigorous diagnosis of LPR or GERD through symptoms, findings, or objective testing. For GERD, the consideration of international criteria was appreciated (Johnson & DeMeester score [16]; Montreal criteria [17]). Patients with positive pH-monitoring or (hypopharyngeal-esophageal) multichannel intraluminal impedance-pH monitoring ((HE) MII-pH) were considered as LPR patients. Patients with reflux esophagitis or positive DeMeester score at the pH-study were considered as GERD patients. Patients with a clinical diagnosis of LPR or GERD without objective testing were considered as 'suspected LPR or GERD patients'.

## Outcomes

The main study outcome was a review of the potential causal association between reflux and the following dental disorders: mucosa inflammation, dental erosion and caries. Gingivitis and periodontitis were considered as mucosa inflammation disorders. The definitions and the tools used for the assessment of these dental disorders are available in Table 1 [18–21]. Authors summarized the following characteristics of studies: the method/criteria used for the reflux diagnosis, the prevalence of dental disorders in the reflux population or the prevalence of reflux in the dental disorder population, the outcome used for the study of association between reflux and dental disorders, and the comparison of outcome with a control-group. The second study outcome was a review of the basic science studies for evaluating ways in which reflux might lead to dental disorders. Heterogeneity among included articles on patient population, means of reflux diagnosis, and outcomes measurements limited ability to combine data statistically into a formal meta-analysis, limiting analysis of the current Systematic Review to qualitative rather than a quantitative summary of the available information. The Tool to Assess Risk of Bias in Cohort Studies developed by the Clarity Group and Evidence Partners was used for the bias/heterogeneity analyses of the included studies [22].

## Intervention and comparison

Because the objective of this systematic review is to analyze the potential relationship between reflux and dental disorders, the included studies did not have to detail treatment approaches or response.

**Table 1. Definitions of dental disorders included in the present review.**

| Disorder | Definition | Assessment tools |
|---|---|---|
| Dental Erosion | The loss of hard dental tissue by a chemical process without bacterial involvement. | Tooth Wear Index (TWI) |
| Tooth decay (dental caries) | The destruction of the outer surface (enamel) of a tooth. | Decayed Missing Filled (DMF) Index |
| Gingivitis | The non-destructive disease causing inflammation of the gums. | Papillary Marginal Attached (PMA) Index |
| Periodontitis | The chronic multifactorial inflammatory disease associated with dysbiotic plaque biofilms and characterized by progressive destruction of the tooth-supporting apparatus. | PMA Index. |

This table presents the scientific definitions of the dental disorders studied in the present review and the clinical scores used in the included papers for their assessment.[18-21] All clinical score/index are characterized by a high score in pathological cases.

## Search strategy

Three independent authors (JRL, CMCE and MRB) conducted a PubMed, Cochrane Library and Scopus search to identify articles published between January 1990 and December 2019 about the role of reflux in the development of dental disorders. There was a high degree of agreement between authors (p<0.05). Clinical and experimental studies were screened if they had database abstracts, available full-texts or titles referring to the condition. The following keywords were used: 'reflux'; 'laryngitis'; 'laryngopharyngeal'; 'gastroesophageal'; 'dental'; 'teeth'; 'decay'; 'caries'; 'erosion'; and 'mucosa'. Authors analyzed the number of patients, study design, inclusion and exclusion criteria, quality of trial and evidence level (EL).

## Results

Initial screening identified 386 papers. Among these papers, 22 articles met our inclusion criteria (Fig 1, Table 2) [11–13,23–41]. Two additional basic research studies were included. These studies investigated the relationship between reflux, dental erosion [42,43] and mucosal inflammation [43].

## Reflux and dental erosion

**Dental erosion in reflux patients.** A total of 18 papers investigated the relationship between reflux and dental erosion in patients with a suspected or confirmed diagnosis of reflux [13,23–34,36,38–41]. The prevalence of dental erosion in reflux patients ranged from 16% to 44%, [27,28,31,34,36,39–41] whereas dental erosion occurred in <20% of healthy individuals [31,36,40]. Overall, controlled studies showed that GERD patients have a significantly higher rate or severity score of dental erosion compared with healthy individuals (Table 2) [13,29–31,36,40].

According to dual-probe esophageal pH testing, Schroeder *et al.* found a higher prevalence of dental erosion in LPR patients (N = 7/10; 70%) compared with GERD patients (N = 3/10; 30%) or controls (N = 1/10; 10%) [24]. In the same vein, Moazez *et al.* reported that patients with pharyngeal acid or weakly acid reflux episodes had higher tooth wear index scores than healthy individuals [12].

About the location of erosions, Gregory-Head *et al.* found that reflux was associated with the development of dental erosion in both mandibular and maxillary surfaces [29]. However, the results of the study of Loffeld *et al.* suggested a differential impact of reflux on the tooth damage [27]. Thus, the rate of upper incisor damage (32.5%) was higher than the rate of lower incisor damage (7.8%) in patients with positive GERD regarding Johnson & DeMeester score. These authors also reported that the duration of GERD complaints was positively associated with the presence of upper incisor damage [27]. Similar findings have been suggested by Filipi *et al.*, who observed that a long GERD history was significantly associated with a higher risk of caries and dental erosion [33].

**Reflux in patients with dental erosion.** Four studies focused on the prevalence of reflux in patients with dental erosion [23,25,26,38]. The prevalence of GERD-symptoms or findings (esophagitis or positive pH testing) in patients with dental erosion ranged from 64% to 75% [25,26,38]. Bartlett *et al.* found that 64% of patients with dental erosion had pathological distal reflux regarding Johnson & DeMeester score [26]. Moreover, they reported that GERD patients with oral acid pH had higher tooth wear index scores. In the same vein, Meurman *et al.* reported that the severity of dental erosion was higher in patients with suspected or confirmed GERD compared with subjects with dental erosion but no GERD [23]. More recently, Wilder-Smith *et al.* did not report a significant association between the characteristics of the distal reflux episodes (acid or weakly acid) and the severity of dental erosion [38].

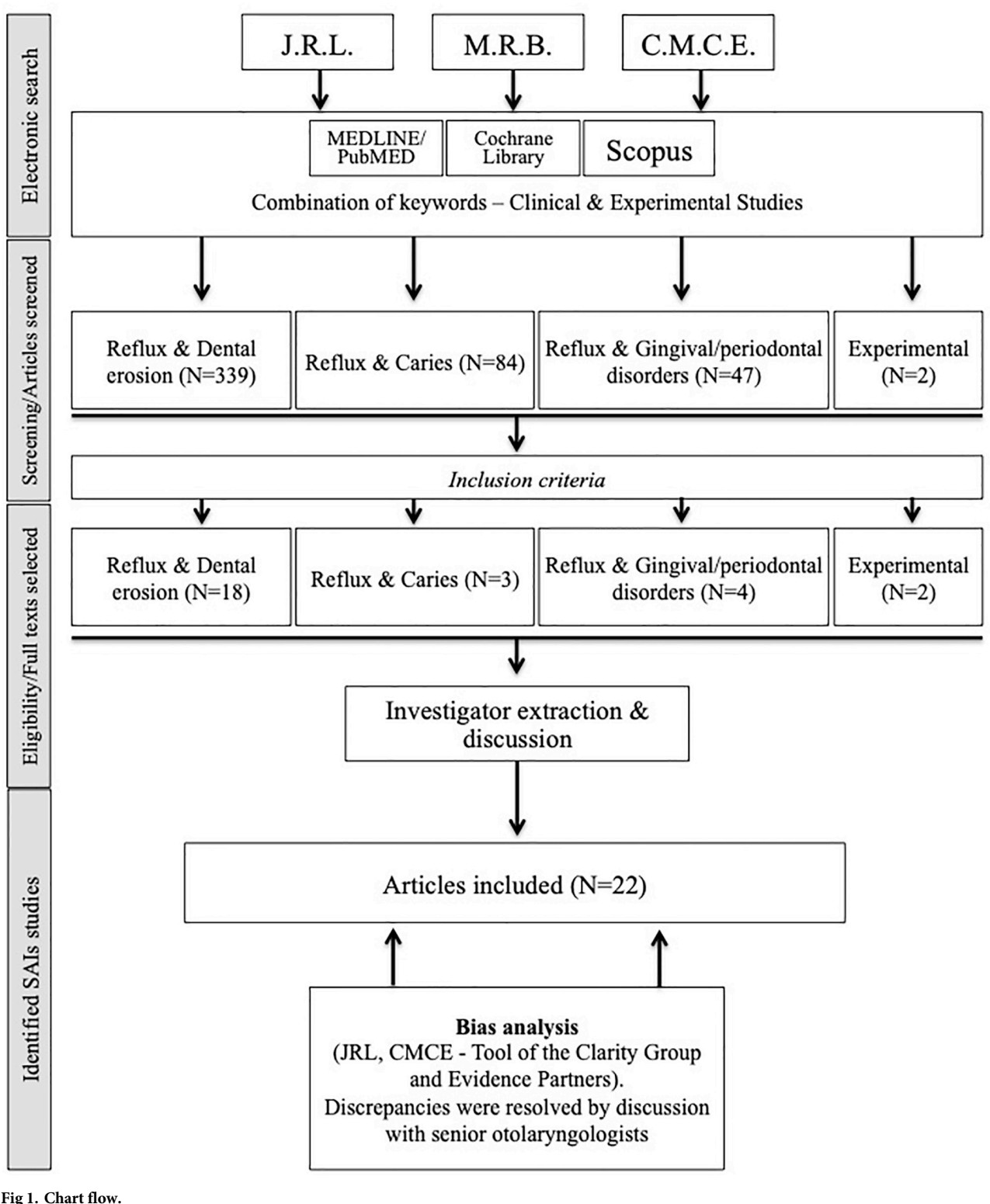

**Fig 1. Chart flow.**

**Table 2. Studies investigating the relationship between reflux and dental disorders.**

| References | Design | EL | Patients characteristics | Reflux diagnosis | Outcome association | Results | Main findings |
|---|---|---|---|---|---|---|---|
| Meurman [23] | Prospective Controlled | B | Gr1: 28 erosion + Gr2: 89 erosion - Gender: 78M/39F Age: 54–49 yo | 1. GERD symptoms 2. In some patients: Single-probe pH study Johnson & DeMeester | Ass. mucosal changes & Reflux hygiene Saliva buffering capacity, viscosity and flow rate. | - Gr1>Gr2 Gr1 = Gr2 Gr1 = Gr2 | Patients with dental erosion have a higher score at the Maratka classification (reflux severity) compared with Individuals without erosion. There are no mucosal and saliva changes associated with reflux. |
| Schroeder [24] | Prospective Controlled | B | Gr1: 12 erosion + Gr2-3: 10 LPR–10 GERD Gr4: 10 CT Gender & Age: N.A. | 1. GERD symptoms 2. Dual-probe pH metry | Prevalance GERD—LPR Saliva: pH, flow rates, buffering capacity, Ca2+ & Phosphorus level Dental erosion (Gr2, 3, 4) | Gr1: N = 9 & 7 Gr1: N = 0 N = 7; 4; 1 | Patients with GERD or LPR at the dual-probe pH metry had a higher proportion of dental erosion. The propor- tion of dental erosion was particularly higher in LPR. There were no saliva disorders associated with reflux. |
| Gudmundsson [25] | Prospective Uncontrolled | C | N = 14 erosion Gender: 12M/2F Age: 8–39 yo | 1. Dual-probe pH metry | Esophagitis GI endoscope (N = 12) | N = 9 | 75% of patients with dental erosion and who benefited from GI endoscopy had esophagitis. |
| Bartlett [26] | Prospective Controlled | B | Gr1: 36 palatal erosion Gr2: 10 CT Gender: N.A. Age: 15–74 yo | 1. GERD symptoms 2. Dual-probe pH metry Johnson & DeMeester | Distal reflux episodes Relation between: Esophageal distal & oral pH TWI & acid oral pH | N = 23 + + | 64% of patients with dental erosions had GERD at the pH monitoring (pH<4, 4% of time). Patients with acid pH in distal esophagus had lower oral pH. The acid oral pH was associated with a higher tooth wear index. |
| Loffeld [27] | Retrospective | C | N = 198 GERD Gender: 118M/80F Age: 17–93 yo | 1. GERD symptoms 2. Reflux esophagitis | Upper incisor damage (%) Lower incisor damage (%) Lower & upper incisor damage (%) | 32.5% 7.8% 26.9% | GERD patients have a higher % of incisor damage. The duration of complaints was positively associated with the presence of upper incisor damage. |
| Jarvinen [28] | Prospective Uncontrolled | C | Gr1: 20 esophagitis + Gr2: 48 esophagitis - Gender & Age: N.A. | 1. GERD symptoms 2. Reflux esophagitis | Dental erosion prevalence (Gr1-2) | N = 4; 0 | 20% of patients with esophagitis have dental erosion. |
| Gregory-Head [29] | Prospective Controlled | B | Gr1: 10 GERD—Gr2: 10 CT Gender: N.A. Age: 18–69 yo | 1. GERD symptoms 2. Dual-probe pH metry Johnson & DeMeester | TWI score Mandibular surface involvement Maxillary surface involvement | GERD>CT GERD>CT GERD>CT | Tooth wear index score was significantly higher in patients with GERD compared with patients with GERD-symptoms and no reflux at the pH monitoring. |
| Munoz [30] | Prospective Controlled | B | Gr1: 181 GERD Gr2: 72 CT Gender: 71M/110F Age: 48 yo | 1. GERD symptoms 2. GI endoscopy 3. Single-probe pH metry | Dental erosion prevalence Periodontal statuts: Plaque index, hemorrhagic index, Gingival recessions | GERD>CT GERD = CT GERD = CT | The % and severity of dental erosion was higher in GERD patients compared with controls. There were no differences between groups about the periodontal status. |
| Moazez, 2005 [12] | Prospective Controlled | B | Gr1: 31 LPR—Gr2: 7 CT Gender: 19M/12F Age: 43 yo | 1. LPR symptoms 2. Dual-probe pH metry | % of distal pH <4 % of pharyngeal pH <5.5 Ass. tooth wear & objective LPR | LPR>CT LPR>CT + | The proportion of palatal tooth wear was higher in patients with a high number of pharyngeal reflux episodes compared with controls. |
| Oginni [31] | Prospective Controlled | B | Gr1: 125 GERD Gr2: 100 CT Gender: 57M/68F Age: 38 yo | 1. GERD symptoms 2. Reflux esophagitis | Impaired TWIs & Gr differences GERD patients CTs | 16%; GERD>CT 5% | Dental erosion is significantly more prevalent in GERD patients compared with CT. |
| Holbrook [32] | Prospective Uncontrolled | B | N = 91 erosion & GERD symptoms Gender & Age: N.A. | 1. GERD symptoms | Low salivary buffering Ass. LSB & erosion Low esophageal pH | 10.4% + 17.7% | 10% of suspected GERD patients had low salivary buffering capacity, which was associated with dental erosion. |
| Filipi [33] | Prospective Uncontrolled | C | N = 24 GERD Gender: 7M/17F Age: 46 yo | 1. GERD symptoms 2. Single-probe pH metry Johnson & DeMeester | Buffering capacity reduction Caries & erosion prevalences | N = 13 long GERD history> short GERD history | Patients with a long history of GERD had a higher risk of caries and erosions compared with short GERD history patients. 54% had buffering capacity disorder. |

(Continued)

**Table 2.** (Continued)

| References | Design | EL | Patients characteristics | Reflux diagnosis | Outcome association | Results | Main findings |
|---|---|---|---|---|---|---|---|
| Correa [13] | Prospective Controlled | B | Gr1 = 30 GERD—Gr2 = 30 CT; Gender: 10M/20F; Age: 33 yo | 1. LPR symptoms 2. Single-probe pH metry 3. GI endoscopy | Dental erosion number & severity; Saliva buffering capacity | GERD>CT; CT>GERD | GERD patients have a higher number (and severity) of dental erosion than CT. The saliva buffering capacity is abnormal in GERD patients. |
| Yoshikawa [34] | Prospective Controlled | B | Gr1 = 40 GERD; Gr2 = 30 CT; Gender: 21M/19F; Age: 69 yo | 1. GERD symptoms; *Montreal criteria* | Decayed missing filled index; Papillary, marginal, attached index; Oral Hygiene Index; Dental erosion prevalence (GERD); Salivary flow volume | GERD>CT; GERD>CT; GERD>CT; 24.3%; CT>GERD | Oral symptoms in GERD are likely to be associated with impaired salivary flow volume. GERD patients have a higher number of decay, erosion, and gingival disorders compared with CT. |
| Song [35] | Retrospective | B | Gr1 = 280 periodontis; Gr2 = 280 CT; Gender: 140M/140F; Age: 49 yo | 1. GERD symptoms; *Montreal criteria* | Ass. GERD & Periodontis; Ass. GERD & caries | +; + | GERD is a risk factor for developing caries and chronic periodontis. |
| Alavi [36] | Prospective Controlled | B | Gr1 = 31 GERD—Gr2 = 71 CT; Gender: N.A.; Age: 30-50 yo | 1. GERD symptoms 2. Reflux esophagitis | GERD dental erosion prevalence; CT dental erosion prevalence; Comparison GERD *vs* CT | 22.6%; 7%; GERD>CT | 22.6% of GERD patients had dental erosion, which was significantly higher compared with CT. |
| Deppe [37] | Prospective Uncontrolled | C | N = 71 GERD/NERD; Gender: 30M/41F; Age: 50 yo | 1. GERD symptoms 2. Reflux esophagitis | Oral mucosa erythema/ulcer; Acidic palatal mucosa lesions; Acidic tongue mucosa lesions | N = 19 (27%) –0 (0%); N = 10 (14%); N = 4 (6%) | 27% of GERD patients have oral mucosa acidic irritation. |
| Wilder-Smith [38] | Prospective Uncontrolled | C | N = 374 erosion +; Gender: 222M/127F; Age: 35 yo | 1. GERD symptoms 2. Acid reflux (MII-pH) | MII-pH GERD prevalence | 69% | 69% of patients with dental erosion had GERD at the MII-pH. There was no significant association between reflux characteristics and dental erosion severity. |
| Vinesh [39] | Prospective Uncontrolled | C | N = 142 GERD; Gender: N.A.; Age: N.A. | 1. GERD symptoms | Dental erosion (%); Periodontis (%); Gingivitis—gingival ulcer (%); Gingival or palatal erythema (%); Mouth floor erythema (%) | 44.0%; 25.5%; 9.9%–2%; 5.7%–2.8%; 1.4% | Dental erosion, periodontitis and gingivitis are the most prevalent dental disorders found in GERD patients. |
| Milani [40] | Prospective Controlled | B | Gr1 = 143 GERD; Gr2 = 274 CT; Gender: 43M/100F; Age: 43 yo | 1. GERD symptoms | GERD dental erosion prevalence; CT dental erosion prevalence; Comparison GERD *vs* CT | 25.9%; 17.2%; GERD>CT | GERD patients had a higher prevalence of dental erosion compared with CT. |
| Watanabe [11] | Retrospective | C | Gr1 = 105 GERD; Gr2 = 50 NER; Gender: 57M/48F; Age: 66 yo | 1. GERD symptoms 2. GI endoscopy: Gr1: reflux esophagitis Gr2: non-erosive reflux | Salivary flow volume; PMA Index scores; OHI-S; Decay indices | CT>GERD; GERD>CT; GERD>CT; GERD>CT | Oral soft tissue disorders, dental erosions and caries are associated with GERD. |
| Warsi [41] | Prospective Uncontrolled | C | N = 187 GERD; Gender: 109M/78F; Age: N.A. | 1. GERD symptoms 2. GI endoscopy 3. Reflux esophagitis | Oral submucous fibrosis (%); Oral ulceration (%); Dental erosion prevalence (%) | 66.3%; 59.4%; 35.3% | 35.3% of GERD patients have dental erosion. Nausea, vomitting, esophagitis, xerostomia, ulcer, gingivitis, & angular cheilitis were associated with GERD. |

Abbreviations: Ass. = association; CT = control(s); EL = evidence level; GERD (s) = gastroesophageal reflux disease (symptoms); GI = gastrointestinal; Gr = group; LPR = laryngopharyngeal reflux; LSB = low salivary buffering; M/F = male/female; MII-pH = multichannel intraluminal impedance pH-monitoring; N.A. = not available; NER = non-erosive reflux; OHI-S = Simplified Oral Hygiene Indice; PMA = Papillary Marginal Attached; TWIs = tooth wear index score; yo = years old.

**Reflux, caries and oral mucosa disorders.** Three studies investigated the relationship between reflux and caries in adults [11,34,35]. Initially, Yoshikawa *et al.* observed higher scores of decayed missing filled index in GERD patients compared with controls [34]. Two years later, Song *et al.* retrospectively supported the positive association between the presence of GERD symptoms and the development of both periodontitis and caries [35]. The results of these two studies were corroborated in the controlled study of Watanabe *et al.*, who found stronger scores of decay indices in GERD patients compared with healthy subjects [11].

The involvement of reflux in the development of both gingival and periodontal disorders has been studied in 5 studies [11,30,37,39,41]. Munoz *et al.* did not find significant differences between patients with positive esophageal pH testing and healthy individuals in the periodontal status, including plaque index, hemorrhagic index, and gingival recessions [30]. More recently, Vinesh *et al.* reported a periodontitis rate of 25.5% in GERD patients [39]. In the same study, the authors reported that <10% of GERD patients had oral mucosa or gingival inflammation findings (erythema, ulcer) [39]. These results did not corroborate those of Yoshikawa *et al.* and Watanabe *et al.* who reported more frequently mucosa inflammatory findings (papillary, marginal, attached index) in GERD patients compared with controls [11,34].

Other mucosal changes have been reported in the study of Deppe *et al.* where 27% of GERD or non-erosive reflux patients had findings suggesting oral irritation (e.g. palatal, buccal and tongue erythema) [37]. The proportion of reflux patients affected by oral mucosal changes was higher in the study of Warsi *et al.*: these authors found that 66.3% and 59.4% of GERD patients had oral submucous fibrosis and oral ulceration(s), respectively [41].

## Potential mechanisms of association

**Clinical studies.** The impact of reflux on the saliva physiology is the most studied field for explaining the relationship between reflux, dental erosion, caries and mucosa inflammation [11,13,23,32–34]. The occurrence of lower salivary buffering capacity in GERD patients has been supported in three studies [14,32,33]. Among the other saliva impairments, studies supported that the low oral pH [32] and the low salivary flow rate [11,34] may be additional factors contributing to the development of dental erosion in GERD patients. Moreover, two studies reported that the oral hygiene index was better in healthy individuals compared with GERD patients [11,34].

**Experimental research.** Higo *et al.* explored the association between dental erosion and GERD in a surgically induced reflux rat model [42]. They observed a significant higher rate of dental erosion, alveolar bone destruction and osteomyelitis in reflux rats compared with controls 30 weeks after the surgery. In a similar rat model, Shimazu *et al.* investigated the development of dental and oropharyngeal lesions [43], in particular pathological changes in the tooth and pharynx on experimental rat model of chronic acid GERD, elucidating the possible association between gastric acid reflux and oral and pharyngeal diseases. The oral cavities were observed histologically every 2 weeks until 20 weeks and the results reported a shorter molar crown heights in GERD rats with dental erosion (10-weeks after) and dentin exposure (20-weeks after), associated with inflammatory cell infiltration of neutrophils and lymphocytes and fibrosis both in the periodontal pocket and in the posterior part of the tongue mucosa.

## Epidemiological analysis

The vast majority of the studies were controlled studies. Overall, 11 studies were prospective controlled (EL: B), 8 were prospective uncontrolled (EL: C), and 3 were retrospective chart reviews (EL: C). There was an important heterogeneity between studies about the reflux

diagnosis approaches. The reflux diagnosis consisted of GERD symptoms ± objective examination(s) in the majority of studies (N = 20).

The reflux diagnosis was based on GERD symptoms and demonstrated reflux esophagitis (N = 7) [11,27,28,31,36,37,41]; GERD symptoms only (N = 5) [32,34,35,39,40]; two other groups using Montreal criteria) [34,35]; or GERD symptoms and positive findings at the single- [30,33] or dual-probe pH monitoring (N = 5) [24,26,29]. Among the authors using the pH monitoring for the diagnosis, 4 authors used Johnson & DeMeester criteria [23,26,29,33], whereas other used composite criteria or did not provide the detailed information for the GERD diagnosis [12,13,24,25,30,38]. Among the composite approaches, Meurman *et al.* recognized using single-probe pH monitoring in some patients of the cohort; the diagnosis of other patients being based on GERD symptoms only [23]. Correa *et al.* included patients with both GERD and LPR symptoms, but the diagnosis confirmation was based on the positive distal acid reflux episodes at the single-probe esophageal pH monitoring [13]. Wilder-Smith *et al.* considered acid and weakly acid reflux episodes through MII-pH findings in patients with GERD symptoms [38]. The detection of acid pharyngeal reflux episodes was considered for the diagnosis of LPR in 2 studies [12,25]. Many authors did not exclude cofactors that may lead to dental disorders, such as bruxism, alcohol or tobacco consumption, medication, radiation and history of dental procedures. The bias analysis is reported in Table 3. For the LPR diagnosis, the following criteria/ratings were considered for the analysis: No = authors based the diagnosis on symptoms or findings only; Probably no = authors based the diagnosis on single probe esophageal pH monitoring or esophagitis; Probably yes = authors based the diagnosis on dual/triple esophageal/pharyngeal probe pH monitoring or pepsin detection in tissue; Yes = authors based the diagnosis on impedance pH monitoring considering acid/nonacid pharyngeal reflux episodes. For the exclusion criteria/confounding factors, the following criteria/ratings were considered for the analysis: No = many conditions were not excluded and the risk of bias in the study results is high; Probably no = some conditions were not excluded and the risk of bias in the study results may be significant; Probably yes = the majority of confounding conditions were excluded and the risk of bias in the study results may be low; Yes = authors carefully excluded the majority of confounding conditions that may bias the interpretation of the study results. For the outcome of association, the following criteria/ratings were considered for the analysis: No = outcomes are not adequate to demonstrate potential association; Probably no = outcomes are less adequate to demonstrate potential association; probably yes = outcomes may be adequate to demonstrate potential association; yes = outcomes are adequate to demonstrate potential association.

## Discussion

Direct and indirect treatment costs due to dental diseases worldwide are estimated from $144 to $442 billion yearly, corresponding to an average of 4.6% of global health expenditure [44]. Dental erosion and caries are among the most prevalent dental diseases, the latter affecting more than 2.5 billion people worldwide [45]. In that context, the identification of favoring factors, such as reflux, makes particularly sense to reduce the considerable economic burden to society related to the management of these disorders.

The main finding of this review is the identification of an important heterogeneity between studies in the method used for the reflux diagnosis. Thus, the large majority of studies considered the reflux diagnosis through GERD criteria (GERD symptoms, reflux esophagitis, esophageal distal reflux episodes) and only two studies really distinguished LPR from GERD [12,25]. Nowadays, there is no consensus about the LPR diagnosis criteria, but many authors agree that LPR may be highly suspected in case of laryngopharyngeal symptoms, findings and ≥1 acid or

**Table 3. Bias analysis.**

| References | LPR diagnosis | Cofactors | Outcomes of association |
|---|---|---|---|
| Meurman [23] | Probably no | Probably yes | Probably yes |
| Schroeder [24] | Probably yes | Yes | Probably yes |
| Gudmundsson [25] | Probably yes | N.A. | No |
| Bartlett [26] | Probably yes | Probably no | Probably no |
| Loffeld [27] | Probably no | N.A. | Probably no |
| Jarvinen [28] | Probably no | N.A. | Probably no |
| Gregory-Head [29] | Probably yes | Probably yes | Probably yes |
| Munoz [30] | Probably no | Probably yes | Probably yes |
| Moazez, 2005 | Probably yes | N.A. | Probably yes |
| Oginni [31] | Probably no | Probably no | Probably no |
| Holbrook [32] | No | Probably yes | Probably no |
| Filipi [33] | Probably no | N.A. | Probably no |
| Correa [13] | Probably no | Probably no | Probably no |
| Yoshikawa [34] | No | Probably no | Probably yes |
| Song [35] | No | Probably no | Probably no |
| Alavi [36] | Probably no | N.A. | No |
| Deppe [37] | Probably no | Probably no | Probably no |
| Wilder-Smith [38] | Probably yes | Yes | Probably yes |
| Vinesh [39] | No | N.A. | Probably no |
| Milani [40] | No | Probably yes | No |
| Watanabe [11] | Probably no | Probably yes | Probably yes |
| Warsi [41] | Probably no | Probably no | Probably no |

The Tool to Assess Risk of Bias in Cohort Studies developed by the Clarity Group and Evidence Partners was used for the bias/heterogeneity analyses of the included studies.[22] Abbreviations: N.A. = not available.

nonacid pharyngeal reflux episodes at the 24-hour HEMII-pH [46–48]. In the majority of studies, the authors did not investigate the occurrence of pharyngeal reflux episodes and did not consider nonacid reflux, which concerns more than 50% of LPR patients [47,48]. The only consideration of reflux diagnosis though GERD criteria is a selection bias because heartburn and other GERD-associated digestive complaints are not present in all LPR patients, and less than 50% of GERD patients have LPR regarding the HEMII-pH [1,49]. For this reason, it is still difficult to drawn a clear conclusion about the impact of LPR in the development of dental disorders. We can state that the prevalence of dental erosion and (to a lesser extent) caries appears to be higher in GERD patients compared with healthy individuals.

The heterogeneity in the patient inclusion criteria may explain the inconsistencies between studies, particularly in the assessment of salivary function. Indeed, some authors identified a significant rate of impaired salivary function in GERD patients [13,32,33], while Meurman *et al.* did not find significant abnormalities [23]. The lack of consideration of pharyngeal reflux episodes is particularly problematic for the analysis of saliva function because to have an impact on the saliva secretion and composition. Reflux has to be characterized by pharyngeal/ oral reflux episodes. The current controversial results found in the literature are probably due to the study of different profiles of patients; some patients having GERD and LPR other subjects having GERD without LPR.

However, the study of the modifications of the saliva function makes sense in the reflux context. Saliva is composed of many protective factors (e.g. epidermal growth factor, mucus,

bicarbonate), which are modified by LPR [50–52]. For example, some works have demonstrated that pepsin negatively impacts the function of carbonic anhydrase type III, which is an essential enzyme for the production of bicarbonate in the pharyngolaryngeal mucosa [53]. Besides, Samuels *et al.* reported that pepsin might impair the expression of different mucin genes, leading to dehydration of the mucus, which becomes less protective [54]. In the same way, LPR is associated with a decrease of epidermal growth factor in the saliva, which may decrease the healing of mucosal lesions [52]. According to these observations and the results of the studies included in the present review, it is reasonable to suspect that reflux may lead to saliva impairments, which may involve flow rate and buffering capacity. The modification of saliva composition and secretion could be associated with a decrease in the dental hygiene status of reflux patients, which is supported in two studies [23,34].

Dental erosion may be related to other etiologies than LPR, including bruxism [55], extrinsic acids (fruit juices, carbonated and isotonic drinks) [56], medication [57], eating disorders [57], alochol and tobacco [30], radiation and history of dental procedures [58]. The epidemiological analysis found that a significant number of authors did not consider these confounding factors in their studies, leading to potential biases.

Strangely, there is an important expanding research area that has not been explored in patients with both reflux and dental disorders: the laryngopharyngeal and oral microbiota. The study of the microbiota is an expanding area in many digestive diseases because it would be associated with the development and the therapeutic response of some inflammatory diseases [59]. In oral and dental disorders, recent studies reported the protective role of some bacteria such as *Akkermansia muciniphila* against the development of periodontal disorders in animal models [60]. Another recent paper supported the pivotal role of oral microbiota in regulating human oral health [61]. According to the characteristics of oral microbiota, some patients would develop more frequently caries than others, due to complex interaction between the commensal microbiota, host susceptibility and environmental factors [62]. Because LPR is associated with laryngopharyngeal and oral pH changes [25,63] and involves the reflux of many digestive enzymes, it is conceivable that the reflux disease may change the local microbiota, which could be associated with modulation of the local inflammation. In the same vein, the consumption of proton pump inhibitors (PPI) is known to be associated with oral microbiota changes [64,65]. Note that, to date, only one study investigated the dental health of GERD patients throughout a PPI clinical course. Wilder-Smith *et al*. found that erosive tooth wear did not progress over the 12-month PPI course [66]. The microbiota of patients with laryngeal carcinoma seems to be modified [67]. To our knowledge, there is no study extensively investigating this field of research. Only two studies suggested potential modification of the prevalence of *Streptococcus mutans* in GERD children [68], but they did not investigate the other commensal microorganisms. Naturally, it is currently impossible to state that LPR induces microbiota changes, but this hypothesis has to be investigated in future experimental and clinical studies. Moreover, the future studies could consider the role of diet in both the oral microbiota changes and the development of reflux. Indeed, both exogenous acids (from the diet) and endogenous acids (from stomach juice) are known to dissolve the enamel mineral, resulting in dental erosions [69].

## Conclusion

The involvement of reflux in the development of dental erosion, caries, and mucosa inflammation is not demonstrated. The lack of use of HEMII-pH, the heterogeneity between studies, and the low level of evidence of studies limit the drawn of clear conclusion. Future clinical controlled studies should consider all types of laryngopharyngeal reflux, the detection of

gastroduodenal proteins in the saliva and the potential interaction between diet, reflux and oral microbiota. The future demonstration of relationship between LPR and oral disorders makes sense regarding the prevalence of both patients with a long history of reflux and dental disorders.

## Supporting information

**S1 Checklist. PRISMA 2009 checklist.**
(DOC)

## Author Contributions

**Conceptualization:** Jerome R. Lechien, Christian Calvo Henriquez, Francois Mouawad, Cyrielle Ristagno, Antonio Schindler, Cyril Bouland, Luigi Laino, Sven Saussez.

**Data curation:** Jerome R. Lechien, Christian Calvo Henriquez, Francois Mouawad, Cyrielle Ristagno, Cyril Bouland.

**Formal analysis:** Jerome R. Lechien, Christian Calvo Henriquez, Francois Mouawad, Andrea Nacci, Cyril Bouland.

**Investigation:** Jerome R. Lechien, Carlos M. Chiesa-Estomba, Christian Calvo Henriquez, Francois Mouawad, Cyrielle Ristagno, Antonio Schindler.

**Methodology:** Jerome R. Lechien, Carlos M. Chiesa-Estomba, Francois Mouawad, Cyrielle Ristagno, Maria Rosaria Barillari, Antonio Schindler, Sven Saussez.

**Supervision:** Jerome R. Lechien, Maria Rosaria Barillari, Andrea Nacci, Luigi Laino, Sven Saussez.

**Validation:** Jerome R. Lechien, Maria Rosaria Barillari, Luigi Laino, Sven Saussez.

**Writing – original draft:** Jerome R. Lechien.

**Writing – review & editing:** Carlos M. Chiesa-Estomba, Christian Calvo Henriquez, Maria Rosaria Barillari, Andrea Nacci, Cyril Bouland, Sven Saussez.

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
