## [Decision Letter · Decision Letter 0]

22 Jul 2020

PONE-D-20-20220

Laryngopharyngeal Reflux, Gastroesophageal Reflux and Dental Disorders: A Systematic Review.

PLOS ONE

Dear Dr. Lechien,

Thank you for submitting your manuscript to PLOS ONE. After careful consideration, we feel that it has merit but does not fully meet PLOS ONE’s publication criteria as it currently stands. Therefore, we invite you to submit a revised version of the manuscript that addresses the points raised during the review process.

We look forward to receiving your revised manuscript.

Kind regards,

Giovanni Cammaroto

Academic Editor

PLOS ONE

Journal Requirements:

2. Please ensure you have included the full electronic search strategy for at least one database and uploaded it as an additional file.

3. We note that you have not included a statement in the Competing Interests or Financial Disclosure section.

We also note that one or more of the authors are employed by a commercial company: Edwards-Realty Enterprise

a. Please provide an complete Funding Statement declaring this commercial affiliation, as well as a statement regarding the Role of Funders in your study. If the funding organization did not play a role in the study design, data collection and analysis, decision to publish, or preparation of the manuscript and only provided financial support in the form of authors' salaries and/or research materials, please review your statements relating to the author contributions, and ensure you have specifically and accurately indicated the role(s) that these authors had in your study. You can update author roles in the Author Contributions section of the online submission form.

b. Please also provide a complete Competing Interests Statement declaring this commercial affiliation along with any other relevant declarations relating to employment, consultancy, patents, products in development, or marketed products, etc.  

Reviewers' comments:

Reviewer's Responses to Questions

**Comments to the Author**

1. Is the manuscript technically sound, and do the data support the conclusions?

Reviewer #1: Yes

2. Has the statistical analysis been performed appropriately and rigorously? 

Reviewer #1: N/A

3. Have the authors made all data underlying the findings in their manuscript fully available?

Reviewer #1: Yes

4. Is the manuscript presented in an intelligible fashion and written in standard English?

Reviewer #1: Yes

5. Review Comments to the Author

Reviewer #1: This is an interesting review regarding laryngopharyngeal reflux, gastroesophageal reflux and dental disorders. No reviews about this specific topic have been published so far. Besides, the study is well conducted and all aspects concerning the topic are analyzed.

Recently, in patients with laryngopharyngeal reflux pepsin has also been identified in tears, please mention this possibility.

Is there any evidence regarding reflux treatment and dental conditions? If they exist, they should be mentioned.

6. PLOS authors have the option to publish the peer review history of their article (what does this mean?). If published, this will include your full peer review and any attached files.

Reviewer #1: No

---

## [Author Response · Author response to Decision Letter 0]

27 Jul 2020

PLOS ONE

Mons, July, 2020

Dear Professor Cammaroto,

I’m sending the revised Systematic Review entitled: “Laryngopharyngeal Reflux, Gastroesophageal Reflux and Dental Disorders: A Systematic Review." (by Lechien et al.).

I undersigned, Jérôme R. Lechien, certify on behalf of all co-authors that the work has not been published previously or submitted elsewhere for review. Any author has financial interest in the subject under discussion. All authors have read and approved the manuscript. Would you be so kind to consider the present manuscript and send us the Reviewer’s comments.

We considered the comments of the reviewer and the editor. 

Reviewer #1: This is an interesting review regarding laryngopharyngeal reflux, gastroesophageal reflux and dental disorders. No reviews about this specific topic have been published so far. Besides, the study is well conducted and all aspects concerning the topic are analyzed.

Thank you. 

Recently, in patients with laryngopharyngeal reflux pepsin has also been identified in tears, please mention this possibility.

We modified the paragraph in the introduction: introduction, line 6 : « Thus, pepsin has been identified in the laryngeal [5], hypopharyngeal [6], oral [7], nasal [7], tears [8], and Eustachian mucosa as well as in the secretion of chronic rhinosinusites9 and chronic media otitis [10]. »

Is there any evidence regarding reflux treatment and dental conditions? If they exist, they should be mentioned.

There is one study, but the data of this study (Wilder-Smith et a., 2017) overlapped those of another study (Wilder-Smith, 2015). 

We included these findings in the discussion : last paragraph of the discussion : « Note that, to date, only one study investigated the dental health of GERD patients throughout a PPI clinical course. Wilder-Smith et al. found that erosive tooth wear did not progress over the 12-month PPI course [66]. »

Best regards,

Jérôme R. LECHIEN, M.D., Ph.D., M.S.

Head and Neck surgery, Laboratory of Anatomy and Cell Biology, Faculty of Medicine, University of Mons (UMONS), Avenue du Champ de mars, 6, B7000 Mons, Belgium, Jerome.Lechien@umons.ac.be

Telephone: +32 65 37 35 84

Fax: +32 65 37 31 42

---

## [Editor Report · Decision Letter 1]

30 Jul 2020

Laryngopharyngeal Reflux, Gastroesophageal Reflux and Dental Disorders: A Systematic Review.

PONE-D-20-20220R1

Dear Dr. Lechien,

We’re pleased to inform you that your manuscript has been judged scientifically suitable for publication and will be formally accepted for publication once it meets all outstanding technical requirements.

Kind regards,

Giovanni Cammaroto

Academic Editor

PLOS ONE
---

## [Editor Report · Acceptance letter]

4 Aug 2020

PONE-D-20-20220R1 

Laryngopharyngeal Reflux, Gastroesophageal Reflux and Dental Disorders: A Systematic Review. 

Dear Dr. Lechien:

I'm pleased to inform you that your manuscript has been deemed suitable for publication in PLOS ONE. Congratulations! Your manuscript is now with our production department. 

Kind regards, 

on behalf of

Dr. Giovanni Cammaroto 

Academic Editor

PLOS ONE